# How Destination City and Source Landholding Factors Influence Migrant Socio-Economic Integration in the Pearl River Delta Metropolitan Region

Xuanyu Liu [1,2], Zehong Wang [3], Yungang Liu [1], Zhigang Zhu [1], Jincan Hu [4], Gao Yang [5] and Yuqu Wang [1,*]

1    School of Geography, South China Normal University, Guangzhou 510631, China; liuxy327@mail2.sysu.edu.cn (X.L.)
2    School of Geography and Planning, Sun Yat-sen University, No. 135, Xingang Xi Road, Guangzhou 510275, China
3    Guangzhou Urban Planning & Design Survey Research Institute, Guangzhou 510060, China; wangzh386@mail2.sysu.edu.cn
4    School of Management, Guangzhou University, Guangzhou 510006, China; hujincan_gz@gzhu.edu.cn
5    School of Cultural Tourism and Geography, Guangdong University of Finance and Economics, Guangzhou 510320, China; lizh327@mail3.sysu.edu.cn
*    Correspondence: wangyq87@mail2.sysu.edu.cn

**Abstract:** Few studies have analyzed the mixed effects of city size and land factors at the macro level on migrant socio-economic integration. On the basis of survey data on migrants in the Pearl River Delta Metropolitan Region (PRDMR), this study developed a system of multidimensional indicators for analyzing the degree of migrant socio-economic integration and factors influencing it. This study demonstrated the following: (1) The overall degree of socio-economic integration of migrants in the PRDMR was low. Factors including city size, hometown landholding, year of birth, education level, gender, and migratory duration exerted effects of varying extents on the degree of the socio-economic integration of migrants. (2) Better job positions were offered and infrastructure was more developed in first-tier cities, so the degree of migrant economic integration was higher, and the sense of identity was stronger in first-tier cities. Given the low housing prices in second-tier cities, migrants therein were more likely to buy a house and achieve family integration, and the degree of their social integration was stronger. (3) In terms of source landholding factors, the degree of socio-economic integration was relatively low among the migrants who owned arable land and homesteads, and who were born outside Guangdong Province. The study tries to measure the socio-economic integration of immigrants more comprehensively and provide reference for the implementation of differentiated socio-economic integration policies and land transfer policies in the immigration and emigration areas.

**Keywords:** migrants; city size; hometown landholding; socio-economic integration; Pearl River Delta Metropolitan Region

## 1. Introduction

The migrant population in China has grown rapidly since the country's reform and opening-up. According to the Seventh National Population Census of the People's Republic of China in 2020, the migrant population reached 376 million, accounting for 26.6% of the total population, with an increase of nearly 70% in 10 years. At present, there is no generally agreed-upon definition of migration. The dictionary of human geography uses "migration" to represent the mobile event of population flow, which must meet two preconditions: it crosses administrative boundaries, and the moving behavior should be maintained for a certain time [1].

By mainly referring from Bulletin of the Seventh National Census by the National Bureau of Statistics and the Office of the Leading Group for the Seventh National Census of

the State Council [2], we defined "migrants (floating population)" as a population having resided in the current city for more than a half year for conducting social and economic activities, with their permanent household registration remaining in another province or another city in the same province. Due to the lack of stable jobs and social security, immigrants cannot enjoy the same social welfare as urban residents. As a result, they often suffer discrimination and become marginalized groups in society [3]. Low "socio-economic integration" of migrants means that they cannot reunite with their families in cities, and a large number of old people, women and children are left behind in rural areas, which brings visible or invisible harm to their physical and mental health [4]. In addition, the contradiction of migrant workers who have settled in the city for a long time but cannot integrate into the city life, a phenomenon known as "semi-urbanization", has also brought a series of challenges to the government's management capability [3].

However, the degrees of migrant socio-economic integration into destination regions are low, with marginalization by the destination society [5], "floating without settling," [6] and a state of "semi-urbanization" accompanying the integration process. In recent years, new forms and characteristics of migration and population flow have been emerging in China, and urbanization is increasingly concentrated in metropolitan regions and urban areas. To ensure that the practical requirements of implementing new urbanization strategies are met in China and the resolution of essential paradoxical situations of the contemporary era is obtained, factors involved in the social integration of the floating population in metropolitan regions warrant investigation.

Migrant socio-economic integration is a key factor promoting the social harmony and stability of a region [7]. Numerous studies have extensively explored migrant socio-economic integration in terms of its definition [8], theoretical implications [9], systems of measurement, and influencing factors (e.g., individual characteristics [10–13], linguistic assimilation [14], social ties [15], and residential segregation [16]), with Giddens' structuration theory as the general basis. Specifically, most scholars have generally considered the objective "structural" constraints of the social and economic integration of the floating population and examined its associations with typical factors, such as household registration system [12], community type [17], and land policy [18].

In recent years, society-oriented, development-oriented, and residence-oriented types of floating populations have gradually increased. With the shifts in industries, urban layouts and spatial patterns in China, the mechanism of the social integration of floating populations has changed, and the process has become increasingly affected by heterogeneities among destination cities [19], particularly in terms of discretion regarding essential public services and livelihood strategies. For example, in large-sized cities, lofty housing prices increase the barriers for migrants to buy a home and achieving family reunification and social integration in the destination region [20], whereas in mid-sized or small-sized cities, despite the lower housing prices, migrants encounter problems such as low wage levels and limited job opportunities. Therefore, the characteristics of and requirements for migrant socio-economic integration vary between cities of different sizes. Although metropolitan regions of China are the main arenas for migration and integration at present and in the future, migrant socio-economic integration in these regions has rarely been the focus of scholarly attention. Particularly, given the rapid growth of labor-intensive industries and migrant populations in metropolitan regions, such as the Pearl River Delta Metropolitan Region (PRDMR), localization-related appeals have become particularly imperative among large foreign populations.

Meanwhile, from the source region factors, whether they have land property in their hometown may also affect the settlement and integration of migrants in the destination city. China's urban–rural dual system stipulates that most rural land belongs to farmers' collective ownership. As stipulated in the Notice of the General Office of the State Council Concerning the Strict Implementation of Laws and Policies on Rural Collectively-Owned Land for Construction (2007), rural residential land is allocated to villagers only. Restricting land transfer provides basic survival guarantees for farmers, but at the same

time, it inevitably leads to farmers' dependence on land, which may make the rural urban migrant more inclined to return to their hometown for settlement, and ultimately resulting in a lower level of economic and social integration for immigrants who own land in their hometown.

Therefore, thoroughly delineating the degree of migrant socio-economic integration in metropolitan regions and analyzing its influencing factors have become the focus for local governments' policy-making, implementation of successful migrant socio-economic integration, and promotion of high-quality urban development in metropolitan regions. Based on the above analysis, we put forward the main research questions of this study: How do destination city and source landholding factors influence subtle socio-economic integration in the Pearl River Delta Metropolitan Region?

The remainder of the paper is organized as follows: Section 2 provides a review of the literature on migrant socio-economic integration as well as the theoretical analysis. Section 3 presents the research data, variables, and methods. Section 4 presents the empirical analysis and results. Finally, Section 5 presents a discussion of results and policy-related recommendations.

## 2. Literature Review

### 2.1. Definition and Estimation of Migrant Socio-Economic Integration

Migrant integration has been an essential research topic in sociology and population geography. The waves of immigration following the Second World War have attracted scholarly attention to the factors that influenced them. For example, Gordon (1964) indicated that social and economic integration involves structural assimilation, cultural exchange, miscegenation, ethnic identity, and social discrimination [9]. Entzinger divided the migrant social integration process into social and economic integration, political integration, cultural integration, and the acceptance or rejection by recipient societies [21]. Kearns and Whitley (2015) further suggested that social and economic integration encompasses social relationship and community awareness, trust and reliance, and safety [22]. The indicators of immigrant integration proposed by the European Union include income, employment, education, health, social inclusion, and active citizenship [23]. Harder constructed a system of indicators of immigrant integration comprising six dimensions, namely psychological, economic, political, social, linguistic, and navigational [24]. Despite the different opinions, socio-economic integration has been increasingly accepted to be a multidimensional concept that involves interaction. In summary, the abovementioned definitions can be divided into objective and subjective aspects. The objective aspect has explicit characteristics and is associated with external social integration, whereas the subjective aspect has implicit characteristics and pertains to internal social integration. The two aspects interact and jointly constitute comprehensive social integration. Studies and current policies have focused on the comprehensive internal or external social integration, or the objective aspect, but multidimensional and systematic studies focusing on subjective social integration are scarce.

Socio-economic integration is considered a multidimensional concept. Therefore, an approach of creating multidimensional indicators for quantitative estimation of the degree of integration has been extensively adopted. However, standardized conceptual connotations, systems of measurement, and evaluation criteria are not available, leading to a certain extent of differences in the results among different case studies. Studies have generally adopted micro-perspective migrant socio-economic integration evaluation systems based on relevant indicators and estimated integration scores in terms of identity shift, economic integration, political participation, and social security [25]. Zhou (2012) suggested that migrant integration involves economic integration, structural integration, social adaptation, cultural adaptation, and identity [26]. Zou et al. (2020) indicated that economic integration, sociocultural integration, and self-identity are the components of socio-economic integration [27]. Wang et al. (2016) suggested that socio-economic integration comprises economic, cultural, and psychological integration and social ties [15]. Similarly, Yang (2015) suggested

that socio-economic integration involves economic integration, social adaptation, cultural acquisition, and psychological identity [8]. Moreover, Lin et al. investigated the topic from the perspectives of economy, social security, integration intention, cultural adaptation, and social interaction [28]. These studies tended to overemphasize the progressive relationships between related dimensions while overlooking the interactions between them. Moreover, the relationships between these dimensions remain controversial. The degree of integration is mainly evaluated through comparison and analysis of the scores for these dimensions and comprehensive scores, along with judgments based on experience; coupling coordination of overall social integration is lacking.

In summary, research on socio-economic integration has gradually shifted from its early focus on indicators for individual dimensions to multidimensional indicators, implying that migrant socio-economic integration is considered a social and economic phenomenon that transforms migrants from foreigners into local citizens. The socio-economic integration process involves a complicated localization process encompassing livelihood/occupation, social identity, living habits, and identity construction, covering external and explicit objective aspects comprising rural–urban transformation in terms of space of production and of living, household registration, and occupational identity, as well as subjective aspects comprising transformation and abandonment in terms of social perception, values, and identity so as to adapt to city life [9,29,30]. Accordingly, migrant socio-economic integration encompasses economic integration, cultural integration, behavioral adaptation, social adaptation, and psychological integration [7,13,31]. Moreover, family migration, an emerging migration trend in recent years [32], has been increasingly adopted by migrants to achieve complete integration [33], but few studies have estimated the degree of family integration.

### 2.2. Factors Influencing the Degree of Migrants' Socio-Economic Integration

The research on the influencing factors of the level of social-economic integration of immigrants can be divided into micro-, meso- and macroscales. Early research focused on the analysis of individual characteristics, mobility, language assimilation, social relations and other meso- and microlevel factors [10]. For example, education, employment status and floating time are positively related to the social and economic integration of immigrants [13,34]. Musgrave (2014) believes that linguistic and cultural diversity will strengthen cross-cultural communication and promote the integration of immigrants. At the same time, married status also positively affects the integration of immigrants [35,36].

With the ever-growing subjectivity and agency of migrants in opting for integration into the destination city, the idea of diverse interactions between destination city, or source region has gradually attracted research attention. Most scholars have observed that migrant integration is a state and a two-way interaction between destination city and source region [37], and is dependent on migrants' preference for and reception of the destination society and the destination society's acceptance of and tolerance for them, whereas migrants' observance and maintenance of their source hometown traditions and the destination society's prejudice and exclusion toward them are factors that lead to segregation. Therefore, the differences in macroscale factors such as in the size and environment of the destination cities, the source landholding and region of origin, deserve our further study.

First, from the destination factors in a macro aspect, regional factors have been increasingly considered as major factors influencing integration. The dimensions of a floating population's social integration process vary between regions. Compared with eastern China, for the floating populations of central and western China, their degrees of integration in terms of economy and psychological identity are lower, but their degrees of integration in terms of social adaptation and cultural acquisition are higher [8]. Liu et al. (2017) investigated the effect of city location factors on the degree of migrant socio-economic integration, and their findings revealed location advantages for socio-economic integration in coastal provinces and municipalities [38]. Given the vast differences in the social, economic, and cultural aspects among cities of different sizes, the difficulty level for migrants

to achieve local integration also varies, and excessively large or small city sizes may be disadvantageous for migrants' integration and citizenization [39]; however, the differences in the degrees of migrant integration in terms of its economic and social dimensions among cities of different sizes in metropolitan regions have rarely been examined.

The success of achieving migrant integration varies from city to city. Related factors are generally closely associated with city size. For example, given the additional job opportunities and better public services in large-sized cities, household registration in these cities is more valuable than that in small-sized ones. Moreover, residing in large-sized cities implies higher cost of living and higher degrees of social exclusion, which increases the likelihood of residential segregation [16]. Accordingly, the degrees of migrant integration in terms of its related dimensions vary by city size [40]. In addition, compared with first-generation migrants, those of newer generations prefer to live in large-sized cities and are more willing to build social networks in the migration destination, thereby facilitating socio-economic integration [41]. Therefore, it is worth further exploring the differences in the socio-economic integration of immigrants under different city sizes.

Secondly, from the perspective of source region factors, land also affects the process of migrant socio-economic integration; however, few studies have analyzed land factors for the degree of migrant socio-economic integration. Studies have revealed that half of the migrants were willing to surrender their hometown land rights for obtaining household registration in cities, and such intention varies among individuals. Therefore, reducing obstacles to rural land transfer boosts the degree of migrant social integration [42,43]. Given China's land system, land serves to ensure that rural people's living conditions are favorable while also preventing them from obtaining corresponding gains from land appreciation through a land withdrawal mechanism. Consequently, rural land impedes the social integration of rural migrants, which may in turn promote the intention among rural migrants who own rural land to return and settle down back home, which reduces their social interaction in the migration destination and the possibility for them to buy a house there, consequently yielding lower degrees of their integration.

With the continual implementation of the rural land reform in recent years, the distribution of China's urban and rural labor markets has gradually become more effective and approximates a migration pattern that increasingly ignores institutional barriers [44]; however, institutional constraints persist. Moreover, land system factors, such as uncertainties regarding rural land property rights and imperfection of the land transfer market, substantially influence decision-making regarding population mobility [45], all of which have a profound impact on population mobility, migration, and integration behavior [46]. Migrants who own rural land are generally rural–urban migrants. Given the urban–rural dual household registration system that has been implemented for a long time in China, the degree of modernization is considerably lower among rural residents than among city residents [47], which impedes migrants from rural areas from integrating into the society of the destination city.

Therefore, we will take the PRDMR in China as the research case, focusing on how destination city factors and source landholding factors influence migrant socio-economic integration in the Pearl River Delta Metropolitan Region. The following part mainly solve the following research problems: First, how can the socio-economic integration level of urban immigrants of different sizes be comprehensively measure by using a multi-dimensional index system? Secondly, from the perspective of destination city factors, how will the employment environment, infrastructure, housing prices and other factors in cities of different sizes within the urban agglomeration affect the level of socio-economic integration of immigrants? Thirdly, from the perspective of landholding factors, how will the ownership of homesteads and arable land in the immigrants' hometown affect the level of socio-economic integration of immigrants? In the following part, we use the entropy method to comprehensively measure the level of socio-economic integration of immigrants and then construct an OLS regression model to discuss the influencing factors behind the level of socio-economic integration in each dimension. By addressing the

above research issues, this paper tries to measure the socio-economic integration of immigrants more comprehensively and provide reference for the implementation of differentiated social-economic integration policies and land transfer policies in the destination and source areas.

### 3. Study Region, Data, and Methods

#### 3.1. Study Region and Data

The study data were obtained through a questionnaire survey conducted by a themed research group comprising researchers from the Sun Yat-sen University, Peking University, and South China Normal University and were collected from migrants in six core cities in the Pearl River Delta (Shenzhen, Guangzhou, Dongguan, Foshan, Zhongshan and Zhuhai) in 2017 (Figure 1). According to the 2017 Ranking of Cities' Business Attractiveness published by Yicai, we classified Shenzhen and Guangzhou as first-tier cities and Dongguan, Foshan, Zhongshan, and Zhuhai as second-tier cities.

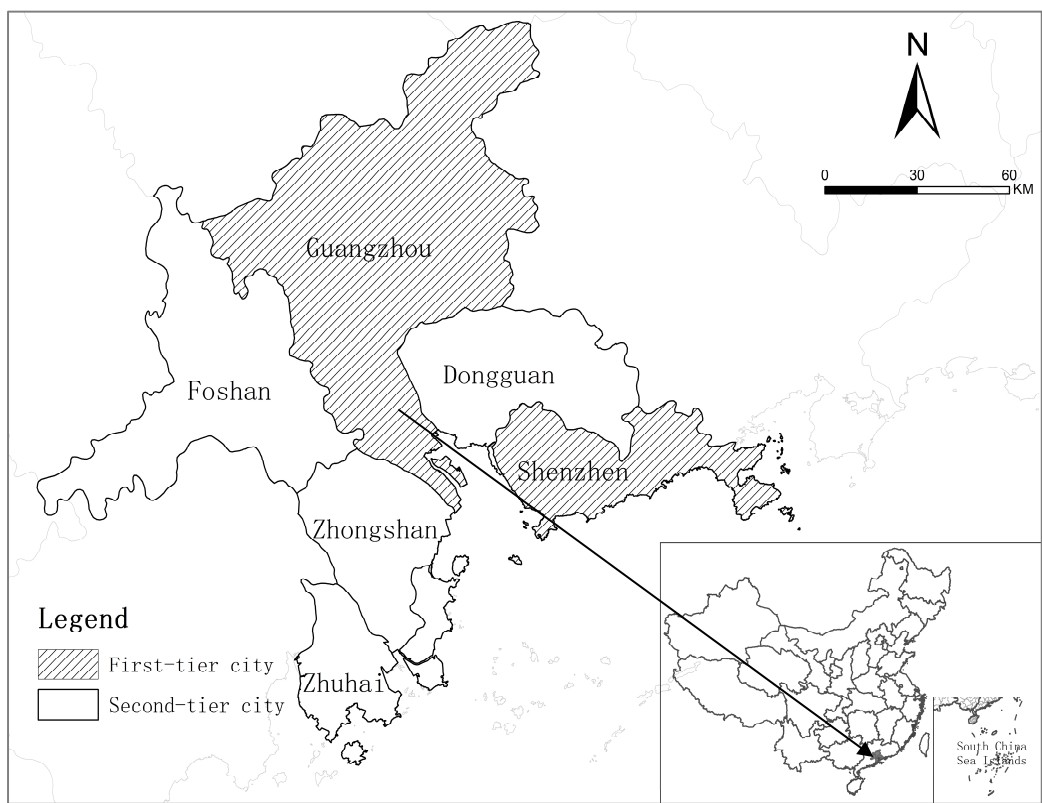

**Figure 1.** Locations of cities in the study region.

We mainly refer to the survey methods and procedures in the National Dynamic Monitoring Survey on Health and Family Planning of Floating Population hosted by the National Health and Family Planning Commission over the years, and use PPS (probability proportional to size) sampling method to investigate the situation of migrants (floating population) [25]. Specifically, based on the 2016 floating population data of each (district- or cluster-district-level administrative region) of the six cities in the Pearl River Delta, a stratified, multi-stage, and scale-proportional PPS method is adopted for sampling. The primary sampling unit is the district; it is to allocate the total sample to the number of survey questionnaires based on the number of immigrants and permanent residents in each district. The number of questionnaires distributed by the research group in each city is as follows: Guangzhou 560; Dongguan, 499; Foshan, 384; Zhongshan, 317; Zhuhai, 293; Shenzhen, 363. Additionally, each district-level administrative district (district) within each

city also uses this as a standard to roughly allocate the number of questionnaires (Figure 1 and Appendix A).

In different district-level administrative regions, we have adopted different specific sampling survey schemes. First, we conduct random surveys in areas where migrants are concentrated, such as regional government units, parks, supermarkets and streets. In addition, we also use the snowball sampling method, which means that the respondents can conduct the survey by recommending no more than five immigrants who meet the survey conditions, but only one respondent is allowed in an immigrant family group. Thirdly, we also adopt a multi-stage survey method, which means that, in order to make the sex, age, education and occupation composition of the respondents in the sample more representative, we organize questionnaire data every week, and then select the respondents in the next stage of investigation to make the composition of the overall sample more representative.

Meanwhile, the individual in-depth interview was carried out on 90 peasant migrants by applying a semi-structured interview guideline. The main interview questions include the migrant population's household registration, employment, labor contract, urban residence, property, children's education, leisure and entertainment. In general, a total of 2416 questionnaires were distributed, and 2358 were obtained for analysis, including 902 from the first-tier cities and 1456 from the second-tier cities.

Among the survey sample (mean age: 34.59 years; mean length of education: 10.64 years), respondents aged 21–50 years constituted the majority. Respondents with household registration in rural areas, male respondents, and respondents born after 1980 constituted the relative majority. Moreover, those with an annual income ranging from RMB 30,000 to 60,000 and those with a junior or senior high school diploma accounted for the largest proportion (Table 1). The survey sample was representative because the demographic characteristics were similar to those of data from previous surveys on migrants in China.

**Table 1.** Demographic characteristics of the respondents.

| Characteristic Index | Number | Percentages (%) |
|---|---|---|
| Female | 1078 | 45.7 |
| Male | 1280 | 54.3 |
| Born in 1980 or before | 831 | 35.2 |
| Born after 1980 | 1527 | 64.8 |
| Unmarried | 789 | 33.5 |
| Married | 1569 | 66.5 |
| Education: College degree below | 1802 | 76.4 |
| Education: College degree or above | 556 | 23.6 |
| Born in Guangdong | 591 | 25.1 |
| Born outside Guangdong | 1767 | 74.9 |
| Annual income: >30 Thousand | 782 | 33.2 |
| Annual income: 30–60 Thousand | 1087 | 46.1 |
| Annual income: 60–100 Thousand | 333 | 14.1 |
| Annual income: >100 Thousand | 156 | 6.6 |
| Rural | 1905 | 80.8 |
| Urban | 453 | 19.2 |

*3.2. Selection of Evaluation Indicators and Value Assignment*

3.2.1. Selection of Indicators

On the basis of relevant studies and the abovementioned definitions, we used an indicator system to assess the degree of socio-economic integration, which comprised four dimensions related to the economic, family, social, and identity aspects. Eleven items in the questionnaire were used as the subindicators of the degree of migrant socio-economic integration (Table 2).

**Table 2.** The indicator system and weight distribution regarding scores of the degree of migrant socio-economic integration.

| Dimensions | Secondary Indexes | Index Weight | Value Assignment |
|---|---|---|---|
| Economic integration | Annual income | 0.0527 | 1 = ($\leq$30); 2 = (30–60); 3 = (60–100); 4 = (>100)(KYuan/1000Yuan) |
| | Occupation | 0.0911 | 1 = Self-employed, private entrepreneurs; 2 = Self-employed, private entrepreneurs; 3 = Professional technicians |
| | Labor contract | 0.0522 | 0 = No; 1 = Yes |
| | Proportion of family reunion in the destination city | 0.0387 | 1 = ($\leq$25%); 2 = (25–50%); 3 = (50–75%); 4 = (75–100%) |
| Family integration Social interaction | Housing ownership in destination city | 0.2369 | 1 = No; 2 = Yes |
| | Frequency of social interaction | 0.0438 | 1 = Never; 2 = Not very often; 3 = Sometimes; 4 = Quite often; 5 = Often |
| | Mastery of local dialect | 0.0420 | 1 = Couldn't understand; 2 = Can understand a little; 3 = Can understand but couldn't speak; 4 = Can speak |
| | City identity | 0.0072 | 1 = Feel excluded; 2 = Don't like or hate, just so-so; 3 = Like it |
| Identity | Household registration | 0.1734 | 1 = Village; 2 = Urban |
| | Settlement intention | 0.0611 | 0 = No; 1 = Yes |
| | Requirements for household registration | 0.2010 | 0 = No; 1 = Yes |

(1) Economic integration: For migrants, achieving socio-economic integration includes migration to a city to conduct nonagricultural activities, making an occupational change, and earning an income that supports their living in the city [27]. The degree of migrant socio-economic integration varies by occupation. Having a high-paying job implies the ability to afford a high level of living in the city [48]. Meanwhile, signing a labor contract implies having a stable job and income [49].

(2) Family integration: Migrants bring their family members to live together in the migration destination, which reflects their stronger intention to permanently reside in the destination city. Family reunion in the destination city also facilitates local social networking and is a major indicator of family integration [50]. For most Chinese people, housing ownership is an essential indicator of a sense of "home" and security for finally attaining family reunion in a city [51]. Moreover, housing ownership is generally associated with family welfare offered in the city, such as a local academic degree and household registration in this city.

(3) Social integration: The frequency of social interaction can indirectly reflect whether migrants can adequately integrate into the destination society. Through interaction with natives, they can form a social identity and attachment and rebuild social networks, thereby increasing the degree of their social integration [52,53]. Understanding the dialect of the migration destination facilitates migrants' social exchange and enhances their sense of belonging [54]. Acceptance is the primary basis for migrants' evaluation of a city, and favorable acceptance indicates a stronger intention of social integration.

(4) Identity: China's household registration system is the main obstacle for migrant socio-economic integration [55]. Household registration in cities entitles migrants to beneficial treatments offered therein, which is a necessary condition for and a major indicator of achieving complete integration. Fulfillment of settlement conditions is necessary for migrants to achieve socio-economic integration, which implies the possibility of having household registration in the destination city. Settlement intention is a psychological indicator of migrant socio-economic integration and is a crucial prerequisite to such integration.

### 3.2.2. Value Assignment and Estimation Method

On the basis of relevant studies, the indicators in each of the four dimensions were assigned equal weights. The subindicators were assigned a value by using the entropy weight method (Table 2). The degree of socio-economic integration was scored using a percentage system. The greater the percentage was, the higher the degree of socio-economic integration was.

### 3.3. Selection of Independent Variables and Model

### 3.3.1. Variable Selection

Indicators of the dimensions of the degree of migrant socio-economic integration were considered dependent variables. Based on previous research and considering the research question, we divided the influencing factors into three aspects: "individual characteristics and immigration experience", "destination city factor" and "source landholding factors," and specifically, destination city factors included city size, healthcare convenience and convenient transportation. Among them, healthcare convenience is used to measure whether it is convenient for immigrants to seek medical resources in the destination city, convenient transportation is used to measure the degree of satisfaction of immigrants with the traffic conditions in the destination city. Source landholding factors included possession of arable land, possession of homesteads, and birthplace. Individual characteristics and immigration experience included gender, year of birth, marital status, education level, and migratory duration (Table 3). An ordinary least square (OLS) regression model was used to examine the effects of the abovementioned independent variables on the degree of migrant socio-economic integration. In order to make the independent variables in the model more comparable, we set all the independent variables to "0" and "1".

**Table 3.** Profile of independent variables and their definitions.

| Factor | Independent Variable | Variable Definitions |
|---|---|---|
| **Individual characteristics and immigration experience** | Gender | 0 = Female; 1 = Male |
| | Year of birth (Whether they are the new generation of migrants) | 0 = After 1980; 1 = Born in 1980 or before |
| | Marital status | 0 = Unmarried; 1 = Married |
| | Education level | 0 = College degree below; 1 = College degree or above |
| | Migratory duration (The total length of time away from hometown) | 0 = ($\leq$10 Year); 1 = (>10 Year) |
| **Destination city factor** | City size (Variable description, see in 3.1) | 0 = Second-tier city; 1 = First-tier city |
| | Healthcare convenience (whether it is convenient for migrants to seek medical resources) | 0 = Inconvenient; 1 = Convenient |
| | Traffic satisfaction (whether immigrants are satisfied with the traffic conditions) | 0 = Dissatisfaction; 1 = satisfaction |
| **Source landholding factors** | Whether they have own arable land | 0 = No; 1 = Yes |
| | Whether they have homestead land | 0 = No; 1 = Yes |
| | Birthplace (Whether they were born in Guangdong province) | 0 = Outside Guangdong province; 1 = Inside Guangdong province |

### 3.3.2. Model

The factors influencing the degree of migrant socio-economic integration were analyzed using the OLS regression model with the following formula:

$$Y = \beta_0 + \sum_{i=1}^{n} \beta_i X_i + \varepsilon_i \tag{1}$$

where Y is the score of the degree of socio-economic integration into a city; $\beta_0$ is the intercept; n is the number of influencing factors; $\beta_i$ is the slope of the *i*-th influencing factor; $X_i$ is the value of the i-th influencing factor; and $\varepsilon_i$ is the error term.

## 4. Results

*4.1. Characteristics of the Scores Exhibited by Migrants in the PRDMR for the Degree of Socio-Economic Integration*

4.1.1. Low Scores for the Degree of Migrant Socio-Economic Integration

The results obtained after value assignment based on the entropy weight method revealed that migrants in the PRDMR had a low score (29.45) for the overall degree of socio-economic integration, which indicated a state somewhat far from complete socio-economic integration. Regarding the dimensions of the degree of integration, the score for the economic integration dimension was low (29.54), indicating that migrants could not obtain higher paying jobs in the migration destination, and their economic income was insufficient. The score for the family integration dimension was 35.91, indicating a low degree of family migration represented by the low degree of urban integration of the migrants' families. Most migrant families did not own a house in the destination city. The scores for the identity (27.06) and social integration (25.31) dimensions were also low, indicating the inadequate integration of the migrants into the destination city, which was mainly evidenced by the high proportion of rural population and unfulfilled settlement conditions. Moreover, considering the added constraints of the household registration system and settlement conditions, identity formation remained a challenge for migrants (Table 4).

**Table 4.** Scores for the degree of migrant socio-economic integration in cities of different sizes.

| Urban Hierarchy | The Overall Degree of Integration | Economic Integration | Family Integration | Social Integration | Identity |
|---|---|---|---|---|---|
| Total | 29.45 | 29.54 | 35.91 | 25.31 | 27.06 |
| First-tier city | 30.05 | 31.85 | 35.03 | 24.97 | 28.35 |
| Second-tier city | 29.09 | 28.11 | 36.45 | 25.53 | 26.26 |

4.1.2. Scores for Migrant Socio-Economic Integration Differed among Cities of Different Sizes

The differences in the scores of migrant socio-economic integration among cities of different sizes reveal that the scores were higher for the first-tier cities (30.05) and slightly lower for the second-tier cities (29.09; Table 4). The scores for the dimensions of socio-economic integration indicate that the migrants in the first-tier cities had higher scores for the economic integration (31.85) and identity (28.35) dimensions in comparison with those in the second-tier cities (28.11 for the economic integration dimension and 26.26 for the identity dimension). However, the migrants in the second-tier cities had higher scores for the family (36.45) and social (25.53) dimensions in comparison with those in the first-tier cities (35.03 for the family dimension and 24.97 for the social dimension).

Regarding specific indicators, the migrants in the first-tier cities exhibited higher scores for five indicators, namely annual income, occupation, settlement intention, household registration, and fulfillment of settlement requirements, in comparison with those in the second-tier cities. Meanwhile, immigrants from second-tier cities scored higher on the four indicators, including social interaction frequency, degree of understanding of the dialect, the proportion of family members cohabiting in the destination city, and housing ownership in destination city, than those in the first-tier cities (Table 5).

**Table 5.** Scores for the indicators of socio-economic integration.

| Dimensions | Variable | Urban Hierarchy | |
|---|---|---|---|
| | | **First-Tier City** | **Second-Tier City** |
| Economic integration | Annual income | 10.89 | 9.82 |
| | Occupation | 17.77 | 15.09 |
| | Labor contract | 3.19 | 3.2 |
| Family integration | Proportion of family reunion in the destination city | 9.48 | 9.98 |
| | Housing ownership in destination city | 25.55 | 26.47 |
| Social integration | Frequency of social interaction | 12.49 | 12.83 |
| | Mastery of local dialect | 10.63 | 10.85 |
| | City identity | 1.85 | 1.85 |
| Identity | Household registration | 21.36 | 20.25 |
| | Settlement intention | 3.56 | 3.36 |
| | Requirements for household registration | 3.43 | 2.65 |

*4.2. Factors Influencing Migrant Socio-Economic Integration*

Before the OLS regression analysis was performed, the independent variables were subjected to a collinearity test. The results revealed that the VIF values were <1.5, indicating the absence of collinearity between these variables. Therefore, they were suitable for OLS regression analysis. In order to test the stability and reliability of the model, we gradually added three aspects to the independent variables, which were "individual characteristics and immunization experience", "destination city factor" and "source landholding factor", to the impact model of social-economic integration indicators in each dimension. The parameters of $\beta_i$ in OLS are generally robust, so the results of the model are reliable overall.

4.2.1. Individual Characteristics and Migration Experience

Gender has significantly affected the scores for immigrant's identity and family dimension scores. For identity dimensions, men had higher scores than women, whereas the opposite result was observed for the family dimension. Women were in a disadvantageous position in terms of income and occupation, but they exhibited relatively high scores for the family integration dimension because migratory women generally move with their husbands. Married migrants exhibited higher scores for the family integration dimension and were in greater need of buying a home in the migration destination. Education level was significantly positively correlated with the scores for all the dimensions. Higher education levels enable the building of stronger human capital so as to obtain a relatively stable job and income, thereby facilitating adaptation to the destination region in terms of behavior, habit, and values and leading to enhanced integration and reduced segregation from the subjective aspect [56].

In terms of age and migratory duration, age significantly positively affected the scores for both the economic integration and social integration dimensions. Migratory duration significantly affected the scores for all the dimensions, and migrants with a longer migratory duration exhibited relatively high scores for the identity dimension. The migrant workers who worked away from their hometown for an extended period have accumulated some material base, human and social capital, which can be better integrated in society. Meanwhile, they are better able to support their families together in the city, making it possible for the whole family to migrate [57].

4.2.2. Destination City Factor

In terms of destination city factor, city size significantly affected migrant socio-economic integration. Given the higher extent of economic development in the first-tier cities, the average income level and job quality there were also better compared with the second-tier cities. The higher income, the greater number of job opportunities, and the

better public service facilities in the first-tier cities enhanced migrants' sense of belonging, all of which led to their higher scores in the economic integration dimension.

We estimated the family integration scores of cities of different sizes. The larger the city size was, the lower the score for the family integration dimension was. The requirement level for permanent residence was lower in second-tier cities in the PRDMR. With the generally much higher income compared to the hometown and the much lower housing prices compared to first-tier cities, the competitive pressure is relatively minor in second-tier cities. These conditions are conducive for local migrants to consider moving their families, and the degree of urban integration of their families was higher [38]. According to the data published in December, 2017, by the China Real Estate Data Academy on the housing prices in the PRDMR, the ratio of unit price per m$^2$ area to income (per month) was 6.84 in Shenzhen and 4.9 in Guangzhou; the ratio in Zhuhai, Dongguan, Foshan and Zhongshan was 3.34, 2.53, 2.17 and 2.01, respectively. The pressure of buying a house was considerably greater in first-tier cities than in second-tier cities. Moreover, city size was negatively correlated with the scores for the social integration dimension, which was probably attributable to the higher cost of living in first-tier cities and higher employee turnover, leading to extended working hours for local migrants and lower frequencies of interaction with natives and resulting in low degrees of social integration. In second-tier cities, given the higher frequency of interaction with natives, the sense of belonging of migrants was stronger [58], and the degree of their social integration was higher. Consequently, migrants in second-tier cities exhibited higher scores for the social integration dimension. Therefore, achieving integration in the social and family dimensions was easier for migrants in second-tier cities than for those in first-tier cities.

The more accessible the healthcare services are in a city, the higher the scores for migrant socio-economic integration are. Healthcare access is an essential indicator of the development of public service facilities in a city and a factor that attracts much attention from migrants. Adequate healthcare facilities increased the social integration scores. In addition, convenient transportation significantly affected migrant socio-economic integration scores. Immigrants who were satisfied with the transport conditions at their destination scored higher on socio-economic integration than the control group. In cities, convenient transportation can help immigrants reduce commuting costs, bring good living experience, and enhance the socio-economic integration of immigrants.

### 4.2.3. Source Landholding Factor

The socio-economic integration scores were relatively low for migrants who owned arable land and homesteads. With the gradual liberalization of the land circulation market, immigrants can get a fixed income by renting contracted land. At the same time, contracted land can guarantee the basic living expenses of returning migrants and improve the attractiveness of migrants' return hometown, which leads to the weakening of the integration motivation of the floating population with contracted land and the phenomenon of low integration [47,59]; however, the effect of homestead was precisely the opposite. Homesteads indirectly affected migrant socio-economic integration scores through the cost effect and income effect. The value of homestead assets and derived housing security has become more prominent under the dual effect of the pursuit of home ownership based on Chinese cultural tradition and the actual dilemma of housing scarcity in the destination region [60], and migrants with the intention to return to their region of origin were less likely to buy a house in and bring their family to the destination city. In particular, for migrants who owned rural homesteads in city suburbs, their expectation of gains from future preservation and appreciation of land in their hometown inevitably and imperceptibly enhanced their identification with their hometown, which reduced the degree of their social integration into the current destination region.

**Table 6.** OLS regression analysis of migrant socio-economic integration.

| | General Integration | | | Economic Integration | | | Family Integration | | | Social Integration | | | Identity | | |
| --- | --- | --- | --- | --- | --- | --- | --- | --- | --- | --- | --- | --- | --- | --- | --- |
| | OLS 1-1 | OLS 1-2 | OLS 1-3 | OLS 2-1 | OLS 2-2 | OLS 2-3 | OLS 3-1 | OLS 3-2 | OLS 3-3 | OLS 4-1 | OLS 4-2 | OLS 4-3 | OLS 5-1 | OLS 5-2 | OLS 5-3 |
| **Individual characteristics and migration experience** | | | | | | | | | | | | | | | |
| Gender (ref: female) | −0.692 *** | −0.585 ** | −0.125 | 0.227 | 0.269 | 0.394 | −1.765 *** | −1.587 *** | −1.308 *** | −1.309 *** | −1.173 *** | −0.279 | 0.080 | 0.154 | 0.695 * |
| Year of birth (ref: after 1980) | 0.991 *** | 0.953 *** | 1.030 *** | 2.322 *** | 2.539 *** | 2.734 *** | 0.269 | 0.169 | 0.283 | 1.762 *** | 1.544 *** | 1.102 ** | −0.387 | −0.439 | 0.000 |
| Marital status (ref: unmarried) | 1.099 *** | 1.038 *** | 1.565 *** | 0.512 | 0.513 | 0.697 | 3.506 *** | 3.501 *** | 3.904 *** | −1.156** | −1.352 *** | −0.373 | 1.532 *** | 1.488 *** | 2.033 *** |
| Education level (ref: college degree below) | 7.796 *** | 8.013 *** | 6.668 *** | 13.060 *** | 12.894 *** | 12.064 *** | 3.079 *** | 3.567 *** | 2.691 *** | 3.857 *** | 4.282 *** | 3.028 *** | 11.188 *** | 11.311 *** | 8.891 *** |
| How long they left home (ref: ≤10 year) | 2.130 *** | 2.094 *** | 2.169 *** | 1.274 *** | 1.268 *** | 1.314 *** | 2.953 *** | 2.925 *** | 2.972 *** | 2.511 *** | 2.428 *** | 2.458 *** | 1.780 *** | 1.754 *** | 1.932 *** |
| **Destination city factor** | | | | | | | | | | | | | | | |
| Urban hierarchy (ref: Second-tier city) | | −0.776 *** | −0.267 | | 0.516 ** | 1.544 *** | | −1.549 *** | −1.544 *** | | −1.711 *** | −1.285 *** | | −0.360 | 0.215 |
| Healthcare convenience (ref: inconvenient) | | 0.198 | 0.376 | | 0.103 | 0.108 | | −0.073 | 0.049 | | 0.516 | 0.972 ** | | 0.246 | 0.375 |
| Convenient transportation (ref: dissatisfaction) | | 0.648 ** | 0.677 ** | | 1.684 *** | 1.700 *** | | 0.418 | 0.457 | | −0.064 | −0.066 | | 0.557 | 0.618 |
| **Source landholding factor** | | | | | | | | | | | | | | | |
| Own arable land (ref: no) | | | −2.254 *** | | | −0.882 ** | | | −0.975 ** | | | −1.823 *** | | | −5.336 *** |
| Have homestead land (ref: no) | | | −3.940 *** | | | −2.949 *** | | | −3.046 *** | | | −0.909 | | | −8.856 *** |
| Birthplace (ref: outside Guangdong Province) | | | 3.259 *** | | | 0.287 | | | 2.727 *** | | | 9.364 *** | | | 0.660 |
| Constant | 27.656 *** | 28.631 *** | 31.034 *** | 25.111 *** | 22.535 *** | 24.483 *** | 35.073 *** | 37.961 *** | 39.040 *** | 25.966 *** | 29.382 *** | 26.315 *** | 24.473 *** | 24.647 *** | 34.297 *** |
| $R^2$ | 0.218 | 0.227 | 0.345 | 0.280 | 0.285 | 0.301 | 0.082 | 0.100 | 0.124 | 0.047 | 0.064 | 0.229 | 0.147 | 0.149 | 0.297 |
| N | 2358 | 2358 | 2358 | 2358 | 2358 | 2358 | 2358 | 2358 | 2358 | 2358 | 2358 | 2358 | 2358 | 2358 | 2358 |
| Variance | 5 | 8 | 11 | 5 | 8 | 11 | 5 | 8 | 11 | 5 | 8 | 11 | 5 | 8 | 11 |

Note: ***, ** and * denote the significance level at 1%, 5% and 10%, respectively.

In terms of birthplace, migrants born in Guangdong Province exhibited a higher score for the social integration dimension compared with those born outside the province. For these migrants born in Guangdong, the situation in the migration destination was similar to that in their region of origin in Guangdong Province with better economic conditions. These migrants were more closely connected to the social network in the migration destination; physically and psychologically more assimilated to "natives"; and more likely to achieve adequate occupational foundation, social relationship, and network rebuilding in the migration destination, which was contradictory to the case of migrants born outside the province. The migrants born in Guangdong Province exhibited higher scores for the identity, economic integration, social integration, and family integration dimensions compared to the migrants born outside the province, confirming that geographic and spatial obstruction influenced migration. For migrants who moved over a shorter distance, higher natural and cultural similarities existed between their region of origin and the destination region, leading to better adaptation to and generally a stronger sense of psychological identity regarding the destination region (Table 6).

## 5. Conclusions and Policy Implications

We analyzed four dimensions of socio-economic integration, namely economy integration, family integration, social integration, and identity on the basis of survey data of migrants in six core cities of different sizes in the PRDMR. An OLS regression model was developed, and factors associated with the scores for each of these dimensions, including destination city and source landholding factors, were explored. The main conclusions of this study are as follows:

The overall scores for migrant socio-economic integration were low, which were affected by mixed effects of macrolevel (e.g., system and policy), mesolevel (e.g., social network and livelihood capital), and microlevel variables (e.g., demographic characteristics and family migration pattern). Migrants who were born after 1980, those who were married and those who had received higher education had relatively high scores in the dimensions of migrant socio-economic integration. Male migrants had higher scores in the economic integration and identity dimensions but lower scores in the family integration dimension. The longer the migratory duration, the higher the score for the identity dimension, but lower the scores for the social integration and family integration dimensions.

In terms of destination city factor, rigid institutional constraints of the household registration system were loosened due to the implementation of flexible policies on equal access to public services, including social security, healthcare services and resources, and transportation facilities, which had a considerable impact on promoting integration and reducing segregation. Therefore, in first-tier cities with greater social security and higher accessibility to basic public services, the migrants had higher scores in the economic integration and identity dimensions compared with migrants in second-tier cities. By contrast, in second-tier cities with low cost of living, the migrants exhibited higher scores in the family integration and social integration dimensions compared with migrants in first-tier cities. The healthcare accessibility in first-tier cities resulted in migrants achieving high scores in the social integration dimension, and migrants who spent a longer time commuting exhibited higher scores for socio-economic integration.

Migrant socio-economic integration scores were also affected by variables such as arable land and homestead. Migrants who owned arable land and homesteads, and those who were born outside of Guangdong Province had relatively low socio-economic integration scores.

The differences in migrant socio-economic integration scores among cities of different sizes in the metropolitan region were analyzed, and the findings revealed that the scores for family integration in cities were lower among migrants in first-tier cities than among those in second-tier cities. Achieving family migration in first-tier cities requires overcoming greater resistance. The analysis by considering the city differences provided a thorough understanding of the characteristics of migrant integration and the integration

process. Generally, previous surveys on influencing factors for migrant integration process considered indicators of only one dimension and regarded the phenomenon of migrant socio-economic integration in cities as an independent event, rather than a multifaceted process, which impeded a clear explanation of migration behaviors [8]. In this study, factors influencing different dimensions of the integration process were compared to sufficiently explore migrants' needs in terms of economy and family integration, which can be used as a reference for the formulation of policies on migrant integration in metropolitan regions. The present study makes the following policy recommendations:

New types of urban construction should be implemented to promote migrant socio-economic integration. First, industrial transformation should be implemented and additional high-paying jobs should be provided to migrants. Moreover, their employability skills should be enhanced through additional career guidance and job training. Second, the provision of public service facilities should be optimized and equalization of basic public services should be accelerated. In addition, the degree of migrants' integration into communities should be promoted, and channels for their involvement in community affairs should be expanded. Third, more attention should be paid to social equity and increasing the scores for socio-economic integration of vulnerable groups, such as women, elderly populations, and people with low education levels.

Differentiated integration policies should be formulated according to the socio-economic integration needs of migrants in different cities, and attempts should be made to increase the overall scores of their socio-economic integration. First, more affordable and public rental housing and housing of shared ownership should be provided in first-tier cities, and it should be ensured that local migrants "have a place to live." Moreover, migrants' social integration should be promoted by creating more sites for social interaction and organizing sports and cultural activities in individual communities. Second, given the relatively ample land resources in the second-tier cities in the Pearl River Delta, buying a house and achieving family migration are less challenging for migrants. Family migration is conducive for migrants to achieve family reunification and solve problems caused by elderly people and children who are left behind [61]. The governments of second-tier cities should create more job positions, increase migrants' income, and provide housing, health-care services, and educational facilities, thereby attracting migrants to move to their cities.

The rural land system should be improved, and a paid withdrawal mechanism should be implemented for migrants' land and homesteads. Moreover, relevant policies for promoting rural land transfer should be formulated to guide migrants to abandon dependence on rural land and enable migrants with a stronger settlement intention to exchange hometown land rights for the capital for buying a house or for the registration indices in the destination city. Therefore, a nationwide paid withdrawal mechanism for migrants' arable land and homesteads should be implemented, and land supply indices of residential land in metropolitan regions should be increased. Furthermore, a unified land market should be established where land indices in the regions of origin may be exchanged for land use indices in the migration destinations, thereby providing affordable housing to most migrants in metropolitan regions.

**Author Contributions:** Conceptualization, Y.W. and Y.L.; methodology, Z.Z.; software, G.Y.; validation, Y.W., Y.L. and Z.W.; formal analysis, X.L.; investigation, Z.W.; resources, Y.L.; data curation, Z.Z.; writing—original draft preparation, X.L.; writing—review and editing, Y.W. and J.H.; visualization, G.Y.; supervision, Z.W.; project administration, Y.L.; funding acquisition, Y.W., J.H. and G.Y. All authors have read and agreed to the published version of the manuscript.

**Funding:** This work was supported by National Natural Science Foundation of China Program-Transnational Migrants Governance in China: based on the Theory of Territoriality, grant number 42071187, by the National Social Science Foundation of China (National Office for philosophy and Social Sciences, 20BSH129 and National Office for philosophy and Social Sciences, 41901194), and by South China Normal University Youth Teacher Research and Training Fund (grant number 22SK07).

**Informed Consent Statement:** Informed consent was obtained from all subjects involved in the study.

**Data Availability Statement:** Not applicable.

**Conflicts of Interest:** The authors declare no conflict of interest.

## Appendix A. Main Characteristics of Migrants in Various Cities and Districts (Cluster) of the Research Area

| District | Net Inflow of Population (10,000 People) | N (Respondents Number) | Average Age | Proportion of Male (%) |
|---|---|---|---|---|
| **Dongguan City** | **627.72** | **499** | **36.32** | **58.52%** |
| Northeast Cluster | 98.89 | 52 | 32.71 | 48.08% |
| Southeast Cluster | 135.85 | 95 | 32.61 | 53.68% |
| Northwest Cluster | 88.78 | 87 | 42.21 | 51.72% |
| Southwest Cluster | 149.7 | 140 | 37.70 | 61.43% |
| Central Cluster | 154.5 | 125 | 34.98 | 68.00% |
| **Foshan City** | **378.68** | **384** | **34.15** | **51.82%** |
| Chancheng District | 61.17 | 56 | 35.95 | 48.21% |
| Gaoming District | 28.85 | 42 | 34.50 | 64.29% |
| Nanhai District | 124.49 | 132 | 32.92 | 53.03% |
| Sanshui District | 38.8 | 23 | 36.52 | 60.87% |
| Shunde District | 125.37 | 131 | 34.10 | 46.56% |
| **Guangzhou City** | **495.98** | **560** | **34.81** | **56.43%** |
| Baiyun District | 148.56 | 77 | 38.26 | 72.73% |
| Conghua District | 1.01 | 25 | 36.92 | 56.00% |
| Panyu District | 68.84 | 84 | 34.63 | 48.81% |
| Haizhu District | 60.32 | 52 | 32.96 | 42.31% |
| Huadu District | 30.9 | 60 | 34.92 | 76.67% |
| Huangpu District | 45.9 | 55 | 30.60 | 49.09% |
| Liwan District | 20.07 | 28 | 37.46 | 71.43% |
| Nansha District | 27.23 | 54 | 32.63 | 68.52% |
| Tianhe District | 70.11 | 72 | 35.61 | 40.28% |
| Yuexiu District | −1.8 | 20 | 34.75 | 40.00% |
| Zengcheng District | 24.78 | 33 | 34.97 | 48.48% |
| **Shenzhen City** | **745.68** | **363** | **32.97** | **53.99%** |
| Bao'an District | 231.52 | 108 | **32.48** | 63.89% |
| Dapeng New Area | 9.06 | | | |
| Futian District | 52.36 | 14 | **33.21** | 57.14% |
| Guangming New Area | 44.25 | 1 | **43.00** | 100.00% |
| Longgang District | 155.03 | 107 | **34.44** | 48.60% |
| Longhua District | 126.95 | 72 | **32.04** | 48.61% |
| Luohu District | 39.46 | 1 | **28.00** | 0.00% |
| Nanshan District | 42.56 | 42 | **32.55** | 47.62% |
| Pingshan District | 28.72 | 18 | **31.33** | 61.11% |
| Yantian District | 15.78 | 0 | | |
| Zhongshan City | 103.98 | **317** | **33.22** | **49.21%** |
| Eastern Cluster | 18.39 | 39 | 36.97 | 48.72% |
| Southern Cluster | 19.63 | 65 | 33.14 | 53.85% |
| Northwest Cluster | 42.64 | 126 | 32.10 | 45.24% |
| Central cluster | 23.32 | 87 | 33.22 | 51.72% |

| District | Net Inflow of Population (10,000 People) | N (Respondents Number) | Average Age | Proportion of Male (%) |
|---|---|---|---|---|
| Zhuhai City | **50.96** | **293** | **35.30** | **50.17%** |
| Doumen District | 7.5 | 73 | 34.08 | 46.58% |
| Jinwan District | 11.98 | 102 | 34.49 | 48.04% |
| Xiangzhou District | 31.48 | 118 | 36.76 | 54.24% |
| Total | | **2416** | **34.59** | **54.06%** |

Note: The net inflow of population is sourced from the statistical yearbooks of various regions.

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
