# Peer review of "How Destination City and Source Landholding Factors Influence Migrant Socio-Economic Integration in the Pearl River Delta Metropolitan Region"

_land, doi:10.3390/land12051073_

Round 1

Reviewer 1 Report

The review paper refers to migrant socioeconomic integration. It was prepared based on the survey which was conducted in Pearl River Delta Metropolitan Region. The findings are interesting however, some elements of the paper have to be changed/ improved.

Most suggestions refer to the introduction.

1. There is no aim of the article formulated.  

2. The author writes: By referring to the literature, we defined “migrants” as a population having resided in the current city for more than a half year for conducting social and economic activities, with their permanent  household registration remaining in a province or another city. Major of definition treat migrant as a person who move to another area, and have stayed there/ live for more than 12 months. Could you give any particular definition that fits the statement – 6 months.  At the same time in theses definition there are information that their permanent  household registration can remain in a province or another city. Could you please enumerate any definition that look at these aspects in that way. I am aware that it is common situation, but I do not recognize any definition which refer to that.

3. The author writes: However, the degrees of migrant socioeconomic integration into host regions are low, with marginalization by the host society, “floating without settling,” and a state of “semi-urbanization” accompanying the integration process. I would suggest changing this sentence and writing it in another way or changing its position – at the end of the paragraph. 

I have some comments on the section: Study Region, Data, and Methods. There is no information about the procedure of the survey, and the contents of the questionnaire. The author should suggest adding a subsection - about the survey procedure.

re

Author Response

Response to comments from the reviewer 1

Many thanks for reviewing the manuscript and giving the feedback. We appreciate the time and effort that you have dedicated to providing the valuable comments, which have revealed the deficiencies that we had before. Please see our response to each of your points below (your comments are in italics).

Comment 1:

The review paper refers to migrant socioeconomic integration. It was prepared based on the survey which was conducted in Pearl River Delta Metropolitan Region. The findings are interesting however, some elements of the paper have to be changed/ improved. Most suggestions refer to the introduction.

Response: We appreciate you reviewed our research and paid so much patience to our manuscript; the suggestions are very helpful for the authors to improve the research. In the revision manuscript, we have modified the introduction, research design, model, results and so on.

Comment 2:

  1. There is no aim of the article formulated.

Response: Thank you for point out the problem. We have added the research aim in the revised manuscript (Page5 Lines 230-246), and revised the related content. The mainly revised content are as follows:

Therefore, we will take the PRDMR in China as the research case, and mainly solve the following research problems: First, how to comprehensively measure the socioeconomic integration level of urban immigrants of different sizes by using the multi-dimensional index system? Secondly, from the perspective of immigration factors, how will the employment environment, infrastructure, housing prices and other factors in cities of different sizes within the urban agglomeration affect the level of socioeconomic integration of immigrants? Thirdly, from the perspective of emigration factors, how will the Ownership of homesteads and arable land of immigrants' hometown affect the level of socioeconomic integration of immigrants? In the following part, we use entropy method to comprehensively measure the level of socioeconomic integration of immigrants, and then construct OLS regression model to discuss the influencing factors behind the level of socioeconomic integration in each dimension. By addressing the above research issues, this paper tries to measure the socioeconomic integration of immigrants more comprehensively, and provide reference for the implementation of differentiated social-economic integration policies and land transfer policies in the immigration and emigration areas.

Comment 3:

  1. The author writes: By referring to the literature, we defined “migrants” as a population having resided in the current city for more than a half year for conducting social and economic activities, with their permanent household registration remaining in a province or another city. Major of definition treat migrant as a person who move to another area, and have stayed there/ live for more than 12 months. Could you give any particular definition that fits the statement – 6 months. At the same time in theses definition there are information that their permanent household registration can remain in a province or another city. Could you please enumerate any definition that look at these aspects in that way. I am aware that it is common situation, but I do not recognize any definition which refer to that.

Response: Thank you for point out the problem and we checked the manuscript and revised the corresponding content. The define of ‘migrants’ in the manuscript is mainly referred from Bulletin of the Seventh National Census (No.7) by National Bureau of Statistics and the Office of the Leading Group for the Seventh National Census of the State Council. We have revise the related content in the revise manuscript (Page 1 Line 37-43).The define of ‘migrants’ in the Bulletin of the Seventh National Census (No.7) is are as follows:

Floating population (‘migrants’) refers to the population separated by households, excluding the population separated by households within the municipal area. The separated population refers to the population whose residence is inconsistent with the township street where the household registration is located and who has left the household registration for more than half a year. The population separated from households in a municipal area refers to the population in the area under the jurisdiction of a municipality directly under the central government or prefecture-level city and between districts, whose residence and household registration are not in the same township street.

Reference: National Bureau of Statistics and the Office of the Leading Group for the Seventh National Census of the State Council. Bulletin of the Seventh National Census (No.7). http://www.gov.cn/guoqing/2021-05/13/content_5606149.htm. (accessed on 18 April 2023).

Comment 4:

  1. The author writes: However, the degrees of migrant socioeconomic integration into host regions are low, with marginalization by the host society, “floating without settling,” and a state of “semi-urbanization” accompanying the integration process. I would suggest changing this sentence and writing it in another way or changing its position – at the end of the paragraph.

Response: Thank you for point out the problem. The revised content are as follows:

In addition, the contradiction of migrant workers who have settled in the city for a long time but cannot integrate into the city life, this phenomenon of "semi-urbanization" has also brought a series of challenges to the government's management capability.

Comment 5:

I have some comments on the section: Study Region, Data, and Methods. There is no information about the procedure of the survey, and the contents of the questionnaire. The author should suggest adding a subsection - about the survey procedure.

Response: We very appreciate your comment here on providing more content in showing on the study areas and surveyed migrants (Page 5-6 Lines 248-270). The revised content are as follows:

The study data were obtained through a questionnaire survey conducted by a themed research group comprising researchers from the Sun Yat-sen University, Peking University, and South China Normal University and were collected from migrants in six core cities in the Pearl River Delta (Shenzhen, Guangzhou, Dongguan, Foshan, Zhongshan, and Zhuhai) in 2017 (Figure 1). According to the 2017 Ranking of Cities’ Business Attractiveness published by Yicai, we classified Shenzhen and Guangzhou as first-tier cities and Dongguan, Foshan, Zhongshan, and Zhuhai as second-tier cities. Surveys were formulated using relevant procedures and local actualities.

In combination with quota sampling, snowball sampling, random sampling, stratified sampling and other methods, the immigrant population in various government department, factories, schools, parks, supermarkets, streets and other areas were investigated. The number of questionnaires distributed by the research group in each city is as follows: Guangzhou 560; Dongguan,499; Foshan,384; Zhongshan,317; Zhuhai, China,293; Shenzhen 363. The individual depth interview was carried out in 70 peasant migrants by applying semi-structured interview guideline. Meanwhile, the individual depth interview was carried out in 90 peasant migrants by applying semi-structured interview guideline. The main interview questions include migrant population's household registration, employment, labor contract, urban residence, property, children's education, leisure and entertainment. In general, a total of 2450 questionnaires were distributed, and 2358 were obtained for analysis, including 902 from the first-tier cities and 1456 from the second-tier cities.

Figure 1. Locations of cities in the study region

We would like to thank you again for all the comments! They have been of great help to the revision of this manuscript! We have tried our best to modify the literature review, results and so on. For your easy processing, we have used the ‘red font track changes’ to revise our manuscript.

Reviewer 2 Report

The content of the abstract can be improved to summarize the main ideas and findings of this study. The findings in the current version of the abstract are inconsistent with those illustrated in the results section.

In the introduction, authors are encouraged to introduce national policies to demonstrate the importance of integrating migrants. It would be better to add some examples to illustrate the problems caused by migrants with low socioeconomic integration.

The type of influencing factors in the literature should be classified, rather than list the influencing factors from each article. It would be better to articulate why the dimensions used in this study are appropriate and what is different from the current study. In addition, the hypotheses in the literature review are too long, so I suggest deleting them. It would be better to substitute the hypotheses with research questions at the end of the literature review.

The variable name should be consistent in this study. For example, the variable name of sex in line 172, could be revised to gender. Urban hierarchy in table 6 could be revised to city size. Contracted land in line 428 could be revised to arable land. Medical care convenience could be revised to health care convenience.

 The data collection process should be detailed. Where and how long did the authors issue the questionnaire in each city, urban village, public rental house or commercial community? The place is relevant to the analysis results. The number of respondents in each city should be described as well.

The description of the birthplace is missing in Table 1. The variable name of year of birth should be added in Table 1.

The format of citations in lines 273 to 275, line 411 and line 503 needs to update.

The measurement of independent variables, such as health care, should also be detailed.

When conducting the OLS regression analysis, we must distinguish discrete and continuous variables. However, the results shown in Table 6 seem that the authors fail to do this. Therefore, I am concerned that the analysis result might be incorrect. Another big concern is that the explanations for the findings are merely from the view of author’s personal deduction, and some explanations conflict with findings shown in Table 6. For instance, when it comes to commute time, it lacks of evidence to illustrate that migrants own a house when commuting for a long time. In addition, the coefficient of arable land is negative, indicating that migrants less actively integrate into the host region, but the explanations in lines 430-433 show the opposite point. The same to lines 382-414. 

Author Response

Response to comments from the reviewer 2

Many thanks for your careful reading and very nice appreciation of this manuscript. Your suggestion on discussing the limitation of the paper is very constructive! We appreciate the time and effort that you have dedicated to providing the valuable comments, which have revealed the deficiencies that we had before. Please see our response to each of your points below (your comments are in italics).

Comment 1:

The content of the abstract can be improved to summarize the main ideas and findings of this study. The findings in the current version of the abstract are inconsistent with those illustrated in the results section.

Response: Many thanks for raising this point. We have proofread and revised part of the abstract (Page1 Lines 14-30). The revised content are as follows:

Abstract: Few studies have analyzed the mixed effects of city size and land factors on different dimensions of migrant socio-economic integration. On the basis of survey data on migrants in the Pearl River Delta Metropolitan Region (PRDMR), this study developed a system of multidimensional indicators for analyzing the degree of migrant socio-economic integration and factors influencing it. This study demonstrated the following: (1) The overall degree of socio-economic integration of migrants in the PRDMR was low. Factors including city size, hometown landholding, year of birth, education level, gender, and migratory duration exerted effects of varied extents on the degree of the socio-economic integration of migrants. (2) The better job positions were offered and infrastructure was more developed in first-tier cities, so the degree of migrant economic integration was higher, and the sense of identity was stronger in first-tier cities. Given the low housing prices in second-tier cities, migrants therein were more likely to buy a house and achieve family integration, and the degree of their social integration was stronger. (3) In terms of hometown landholding and region of origin, the degree of socio-economic integration was relatively low among the migrants who owned arable land and homestead and who were born outside Guangdong Province. The study tries to measure the socio-economic integration of immigrants more comprehensively, and provide reference for the implementation of differentiated socio-economic integration policies and land transfer policies in the immigration and emigration areas.

Comment 2:

In the introduction, authors are encouraged to introduce national policies to demonstrate the importance of integrating migrants. It would be better to add some examples to illustrate the problems caused by migrants with low socioeconomic integration.

Response: We are very grateful for your suggestion here! The examples to illustrate the problems caused by migrants with low socioeconomic integration has been added in the research design section (Page 1-2 Lines 43-51). The revised content are as follows:

Due to the lack of stable jobs and social security, immigrants cannot enjoy the same social welfare as urban residents. As a result, they often suffer discrimination and be-come marginalized groups in society. Low 'socio-economic integration' of migrants means that they cannot reunite with their families in cities and a large number of old people, women and children are left behind in rural areas, which brings visible or in-visible harm to their physical and mental health. In addition, the contradiction of migrant workers who have settled in the city for a long time but cannot integrate into the city life, this phenomenon of "semi-urbanization" has also brought a series of challenges to the government's management capability.

Reference

Bai Y, Yang N, Wang L, et al. The impacts of maternal migration on the cognitive development of preschool-aged children left behind in rural China[J]. World Development, 2022, 158: 106007.

Wu Y, Sun X, Sun L, et al. Optimizing the governance model of urban villages based on integration of inclu-siveness and urban service boundary (USB): A Chinese case study[J]. Cities, 2020, 96: 102427.

Comment 3:

The type of influencing factors in the literature should be classified, rather than list the influencing factors from each article. It would be better to articulate why the dimensions used in this study are appropriate and what is different from the current study. In addition, the hypotheses in the literature review are too long, so I suggest deleting them. It would be better to substitute the hypotheses with research questions at the end of the literature review.

Comment 3-1:The type of influencing factors in the literature should be classified, rather than list the influencing factors from each article. It would be better to articulate why the dimensions used in this study are appropriate and what is different from the current study.

Response: Many thanks for raising this point. We have proofread and revised part of the “2.2 Factors Influencing the Degree of Migrants’ Socio-economic Integration”, and the revised content can be seen in Page 4 Lines 159-202.

Comment 3-2:In addition, the hypotheses in the literature review are too long, so I suggest deleting them. It would be better to substitute the hypotheses with research questions at the end of the literature review.

Response: Thank you for point out the problem. We have substituted the hypotheses with research questions at the end of the literature review in the revised manuscript (Page 5 Lines 230-246), and the revised content are as follows:

Therefore, we will take the PRDMR in China as the research case, focusing on the macro-level impact of immigration and emigration factors on the socio-economic integration of immigrants,  and mainly solve the following research problems: First, how to comprehensively measure the socioeconomic integration level of urban immigrants of different sizes by using the multi-dimensional index system? Secondly, from the perspective of immigration factors, how will the employment environment, infrastructure, housing prices and other factors in cities of different sizes within the urban agglomeration affect the level of socioeconomic integration of immigrants? Thirdly, from the perspective of emigration factors, how will the Ownership of homesteads and arable land of immigrants' hometown affect the level of socioeconomic integration of immigrants?

Comment 4:

The variable name should be consistent in this study. For example, the variable name of sex in line 172, could be revised to gender. Urban hierarchy in table 6 could be revised to city size. Contracted land in line 428 could be revised to arable land. Medical care convenience could be revised to health care convenience.

Response: Thank you for point out the problem! We checked the manuscript and revised the corresponding content. Unified variables are displayed in red font in the revised manuscript.

Comment 5:

The data collection process should be detailed. Where and how long did the authors issue the questionnaire in each city, urban village, public rental house or commercial community? The place is relevant to the analysis results. The number of respondents in each city should be described as well.

Response: We very appreciate your comment here on providing more content in showing on the study areas and surveyed migrants (Page 5-6 Lines 249-267). The revised content are as follows:

The study data were obtained through a questionnaire survey conducted by a themed research group comprising researchers from the Sun Yat-sen University, Peking University, and South China Normal University and were collected from migrants in six core cities in the Pearl River Delta (Shenzhen, Guangzhou, Dongguan, Foshan, Zhongshan, and Zhuhai) in 2017 (Figure 1). According to the 2017 Ranking of Cities’ Business Attractiveness published by Yicai, we classified Shenzhen and Guangzhou as first-tier cities and Dongguan, Foshan, Zhongshan, and Zhuhai as second-tier cities. Surveys were formulated using relevant procedures and local actualities.

In combination with quota sampling, snowball sampling, random sampling, stratified sampling and other methods, the immigrant population in various government department, factories, schools, parks, supermarkets, streets and other areas were investigated. The number of questionnaires distributed by the research group in each city is as follows: Guangzhou 560; Dongguan,499; Foshan,384; Zhongshan,317; Zhuhai, China,293; Shenzhen 363. The individual depth interview was carried out in 70 peasant migrants by applying semi-structured interview guideline. Meanwhile, the individual depth interview was carried out in 90 peasant migrants by applying semi-structured interview guideline. The main interview questions include migrant population's household registration, employment, labor contract, urban residence, property, children's education, leisure and entertainment. In general, a total of 2450 questionnaires were distributed, and 2358 were obtained for analysis, including 902 from the first-tier cities and 1456 from the second-tier cities (Figure 1.).

Comment 6:

The description of the birthplace is missing in Table 1. The variable name of year of birth should be added in Table 1.

Response: Thank you very much for your suggestion. We add the birthplace and the name of year of birth in Table 1.

Comment 7:

The format of citations in lines 273 to 275, line 411 and line 503 needs to update.

Response: Many thanks for reminding this point! We have updated the format of citations in the revised manuscript.

Comment 8:

The measurement of independent variables, such as health care, should also be detailed.

Response: We are very grateful for your suggestion here! We have rewritten the related content (Page 9 Lines 326-340):

Indicators of the dimensions of the degree of migrant socio-economic integration were considered dependent variables. Based on previous research and considering the research question, we divided the influencing factors into three aspects: “individual characteristics and immigration experience”, “city size and infrastructure” and “hometown landholding and region of origin,” and specifically, city size and infrastructure included city size, health care convenience, and convenient transportation. Among them, health care convenience is used to measure whether it is convenient for immigrants to seek medical resources in the destination city, convenient transportation is used to measure the degree of satisfaction of immigrants with the traffic conditions in the destination city. Hometown landholding and region of origin included possession of arable land, possession of homesteads, and birthplace. Individual characteristics and immigration experience included gender, year of birth, marital status, education level, and migratory duration (Table 3). An ordinary least square (OLS) regression model was used to examine the effects of the abovementioned independent variables on the degree of migrant socio-economic integration. In order to make the independent variables in the model more comparable, we set all the independent variables to 0, 1.

Comment 9:

When conducting the OLS regression analysis, we must distinguish discrete and continuous variables. However, the results shown in Table 6 seem that the authors fail to do this. Therefore, I am concerned that the analysis result might be incorrect. Another big concern is that the explanations for the findings are merely from the view of author’s personal deduction, and some explanations conflict with findings shown in Table 6. For instance, when it comes to commute time, it lacks of evidence to illustrate that migrants own a house when commuting for a long time. In addition, the coefficient of arable land is negative, indicating that migrants less actively integrate into the host region, but the explanations in lines 430-433 show the opposite point. The same to lines 382-414.

Comment 3-1:When conducting the OLS regression analysis, we must distinguish discrete and continuous variables. However, the results shown in Table 6 seem that the authors fail to do this. Therefore, I am concerned that the analysis result might be incorrect.

Response: Many thanks for raising this point. In order to improve and test the stability and reliability of the model, we have unified the independent variables and improved the model by gradually adding independent variables, including the following two points:

Firstly, in order to make the independent variables in the model more comparable, we set all the independent variables to “0”, “1” (See in Table 3 in the revised manuscript).

Secondly, we gradually add three aspects independent variables in the OLS regression model, including "individual characteristics and immunization experience", “city size and infrastructure” and “hometown land-holding and region of origin”, to the impact model of social-economic integration indicators in each dimension. The parameters of βi in OLS presented in Table 6 are robust generally, so the results of the model are reliable overall.

Comment 3-2:Another big concern is that the explanations for the findings are merely from the view of author’s personal deduction, and some explanations conflict with findings shown in Table 6. For instance, when it comes to commute time, it lacks of evidence to illustrate that migrants own a house when commuting for a long time. In addition, the coefficient of arable land is negative, indicating that migrants less actively integrate into the host region, but the explanations in lines 430-433 show the opposite point. The same to lines 382-414.

Response: Thank you very much for your suggestion. First of all, we condensed the relevant content and deleted some explanations from the perspective of personal inference. Secondly, we deleted the variable Commute time and redesigned the model. Thirdly, we proofread and improved the original result analysis, combined with the new model results, in order to make the new results more reasonable. The mainly revised content can be seen in Page 11-12, Line391-488.

We would like to thank you again, very sincerely, for all the comments. They have not only helped us improve the paper itself but also given us huge inspiration on future work. We have tried our best to modify the literature review, model, results and so on. We hope we have adequately responded to the points that you raised. Due to the time limit and other reason, there are may still some parts of the manuscript that have not been modified completely. If there are other problems with the manuscript, we hope to have the opportunity to further revise and look forward to hearing from you! For your easy processing, we have used the ‘red font track changes’ to revise our manuscript.

Round 2

Reviewer 1 Report

The paper develops a system of multidimensional indicators for analyzing the degree of migrant socioeconomic integration and the factors influencing it. I have still some comments on the introduction and the section which refers to the methodology. 

1. There is still an international definition of migrants missing. I understand that authors analyze the situation in China, however, there should be more common definitions presented - for example IMO, an adaptation of UN definitions. 

2. There is still a clearly defined purpose of the article missing. 

3. Authors write "In combination with quota sampling, snowball sampling, random sampling, stratified sampling, and other methods, the immigrant population in various government departments, factories, schools, parks, supermarkets, streets, and other areas were investigated". It seems like not planned research but like randomly handing out survey questionnaires to respondents. There is information that it was quota sampling - based on what data? 

Author Response

Response to comments from the reviewer 1

Many thanks for your careful reading of this manuscript. Your suggestion on discussing the limitation of the paper is very constructive! We appreciate the time and effort that you have dedicated to providing the valuable comments, which have revealed the deficiencies that we had before. Please see our response to each of your points below (your comments are in italics).

Comment 1:

There is still an international definition of migrants missing. I understand that authors analyze the situation in China, however, there should be more common definitions presented - for example IMO, an adaptation of UN definitions. 

Response: Many thanks for raising this point. We have proofread and revised part of definitions, and added more common definitions presented - for example IMO, an adaptation of UN definitions (Page1-2 Lines 39-68). The revised content are as follows:

At present, there is no generally agreed definition of migration. The dictionary of hu-man geography uses "migration" to represent the mobile event of population flow, which must meet two preconditions: (1) it crosses a certain administrative boundary; (2) The moving behavior should be maintained for a certain time [1]. In addition to the general definitions of immigrants and immigrants in the dictionary, there are specific definitions of various key terms related to immigration, including definitions in the fields of law, administration, research and statistics. Traditionally, immigrants are classified according to four criteria: domestic and international; temporary and per-manent; forced and voluntary; legal and illegal [2]. The main difference between the concepts of international migration and domestic migration lies in whether individuals move across international borders [1].

In 1998, in order to collect migration data, the United Nations Department of Economic and Social Affairs (UN-DESA) defined "international migration" as "anyone who changes his usual country of residence". The definition of the UN Department of Economic and Social Affairs does not include the mobility caused by "entertainment, vacation, visiting relatives and friends, business, medical treatment or religious pil-grimage" [3]. The International Organization for Migration (IOM) puts forward a con-ceptual framework, which links international immigrants with the existing population, excluding residents temporarily living abroad. The key factor that distinguishes inter-national migrants from other international population movements is the time they stay at home or abroad. Therefore, the standard for defining an immigrant should be the time required for the person to be regarded as a part of the permanent population. In practice, this means that the duration is 6 months or 12 months, so that the flows of migration can be consistent with the annual change of resident population [4].

By mainly referring from Bulletin of the Seventh National Census by National Bureau of Statistics and the Office of the Leading Group for the Seventh National Census of the State Council [5], we defined “migrants (floating population)” as a pop-ulation having resided in the current city for more than a half year for conducting social and economic activities, with their permanent household registration remaining in an-other province or another city in the same province.

Reference

1.Johnston R J, Gregory D R, Pratt G, et al. The dictionary of human geography. Wiley-Blackwell, 2009.

  1. Bailey A J. Turning transnational: notes on the theorisation of international migration. International Journal of Population Geography, 2001, 7(6): 413-428.
  2. Sironi, A. C. Bauloz and M. Emmanuel (eds.), 2019. Glossary on Migration. International Migration Law, No. 34. Interna-tional Organization for Migration (IOM), Geneva.
  3. McAuliffe, M. and A. Triandafyllidou (eds.), 2021. World Migration Report 2022. International Organization for Migration (IOM), Geneva.
  4. National Bureau of Statistics. Bulletin of the Seventh National Census (No.7). http://www.gov.cn/guoqing/2021-05/13/content_5606149.htm. (accessed on 18 April 2023).

Comment 2:

There is still a clearly defined purpose of the article missing. 

Response: Many thanks for raising this point. First of all, we further condensed the scientific questions and revised the title of the paper accordingly: How Destination City, Source Landholding Factors Influence Migrant Socio-economic Integration in the Pearl River Delta Metropolitan Region?

 Secondly, based on the title, we focus on the impact of macro scale factors such as the place of departure and the place of entry on the socio-economic integration of immigrants, and based on this, we will reorganize the literature. Based on the topic, we focus on the influence of macro-scale factors, such as the destination city factor and source landholding related factors, on the socio-economic integration of immigrants, and reorganize the literature accordingly. The mainly revised content can be seen in the ‘red font track changes’ in Page 1-5, Line 1-249.

Thirdly, according to the new title and revised literature review, we have revised the research aim in the revised manuscript (Page5 Lines 256-269), and revised the related content. The mainly revised content are as follows:

Therefore, we will take the PRDMR in China as the research case, focusing on how the destination city factor and source landholding factor influence migrant socio-economic integration in the Pearl River Delta Metropolitan Region. The following part mainly solve the following research problems: First, how to comprehensively measure the socio-economic integration level of urban immigrants of different sizes by using the multi-dimensional index system? Secondly, from the perspective of destination city factors, how will the employment environment, infrastructure, housing prices and other factors in cities of different sizes within the urban agglomeration affect the level of socio-economic integration of immigrants? Thirdly, from the perspective of landholding factors, how will the Ownership of homesteads and arable land of immigrants' hometown affect the level of socio-economic integration of immigrants?

Comment 3:

Authors write "In combination with quota sampling, snowball sampling, random sampling, stratified sampling, and other methods, the immigrant population in various government departments, factories, schools, parks, supermarkets, streets, and other areas were investigated". It seems like not planned research but like randomly handing out survey questionnaires to respondents. There is information that it was quota sampling - based on what data? 

Response: Thank you very much for your suggestion. We have proofread and revised part of the Study Region and Data (Page1 Lines 276-302). At the same time, our research involves planned stratified random sampling. Some of our research subjects were randomly selected, but our research area and proportion of research subjects were determined based on the actual composition of the migrants. The following is a detailed description of our Study Region and Data:

The data for this article were derived from a joint project survey involving quantitative questionnaires and qualitative interviews conducted in the Pearl River Delta Metropolitan Region in Guangdong Province from May to October, 2017. We conducted a survey in six cities in the core area of the Pearl River Delta Metropolitan Region and included scholars and graduate students from various universities and research institutes. According to the size of the city population, net inflow of population, and operability of the survey, the research group roughly allocated the number of questionnaires for each city—Guangzhou: 560, Dongguan: 499, Foshan: 384, Zhongshan: 317, Zhuhai: 293, and Shenzhen: 363. And each district level administrative district (district or cluster) within each city also uses this as a standard to roughly allocate the number of questionnaires. In each city, multistage stratified sampling was used to select individuals. Crucially, the number and distribution of the sample in each city and district was controlled according to the total number and distribution of the migrant population (Fig. 1). In general, a total of 2416 questionnaires were distributed, and 2358 were obtained for analysis, including 902 from the first-tier cities and 1456 from the second-tier cities.

All the interviewees were members of the migrants (floating population) who did not have registered households in the urban area of the city [had households registered in other counties or cities (whether rural or urban)] and who worked and lived in the research area for more than 6 months. The survey mainly combined quota sampling, snowball sampling introduced by acquaintances, random sampling, and stratified sampling method to investigate the migrants in various regional government units, factories, parks, supermarkets, and streets. Investigators conducted face-to-face interviews. To ensure the representativeness of the questionnaire, the number of questionnaires in each survey point should not exceed 15.

The survey mainly combined quota sampling, snowball sampling introduced by acquaintances, random sampling, and stratified sampling method to investigate the floating population in various regional government units, factories, parks, supermarkets, and streets. Investigators conducted face-to-face interviews. Each interview lasted approximately 50 minutes. The survey mainly collected information regarding the personal and family situation of the floating population, including the household registration, employment, labor contract, property, and education of children. To improve the reliability and robustness of the questionnaire, not more than 15 questionnaires were usually given at a specific area. Moreover, we provided remuneration to the respondents and registered their telephone for verification. We primarily used descriptive statistics and a logistic regression model to analyze the data from the 2,416 cases.

Among the survey sample (mean age: 34.59 years; mean length of education: 10.64 years), respondents aged 21–50 years constituted the majority. Respondents with household registration in rural areas, male respondents, and respondents born after 1980 constituted the relative majority. Moreover, those with an annual income ranging from RMB 30,000 to 60,000 and those with a junior or senior high school diploma ac-counted for the largest proportion (Table 1, Annex 1. in the manuscript). The survey sample was representative because the demographic characteristics were similar to those of data from previous surveys on migrants in China.

Tab.1 Main Characteristics of Migrants in Various Cities and Districts (Cluster) of the Research Area

We would like to thank you again, very sincerely, for all the comments. They have not only helped us improve the paper itself but also given us huge inspiration on future work. We have tried our best to modify the literature review and so on. We hope we have adequately responded to the points that you raised. Due to the time limit and other reason, there are may still some parts of the manuscript that have not been modified completely. If there are other problems with the manuscript, we hope to have the opportunity to further revise and look forward to hearing from you! For your easy processing, we have used the ‘red font track changes’ to revise our manuscript.

Reviewer 2 Report

The authors need to set the reference value for the discrete variables in table 6.

Author Response

Response to comments from the reviewer 2

Many thanks for your valuable comments! which have revealed the deficiencies that we had before. Please see our response to each of your points below (your comments are in italics).

Comment 1:

The authors need to set the reference value for the discrete variables in table 6.

Response: Many thanks for raising this point. We have set the reference value for the discrete variables in Table 6.
